# Early postoperative alignment and preoperative stereopsis as determinants of success in intermittent exotropia surgery

**Worapot Srimanan** [ORCID]**\*, Phawasutthi Keokajee**

Division of Ophthalmology, Phramon Hospital, Bangkok, Thailand

* drworapotsmn@gmail.com

## Abstract

To assess the surgical outcomes of patients with intermittent exotropia (IXT) and identify factors influencing surgical success in a tertiary hospital setting. This retrospective study included 150 patients who underwent surgical correction for IXT of less than 50 prism diopters between 2013 and 2024. Stereopsis was evaluated using the Randot stereo test. Surgical success was defined as residual exotropia <10 prism diopters without diplopia symptoms. Preoperative variables were analyzed to determine predictors of surgical success. Surgical success was achieved in 74% (111/150) of patients. The sensory status at one year was 83.46% (111/133). The procedures performed were bilateral lateral rectus recession (56%) and unilateral recess-resection (44%). Preoperative stereopsis and postoperative overcorrection at one week were significantly associated with successful outcomes, with adjusted odds ratios of 2.81 (95% CI, 1.14–6.93) and 4.56 (95% CI, 1.60–12.96), respectively. Reoperations were required in 7.3% (11/150) of cases. Favorable ocular alignment was sustained postoperatively during the first six months but gradually declined over time. Surgical management of IXT demonstrated good motor outcomes, with preoperative stereopsis and postoperative at 1 week overcorrection identified as key factors for success. Further prospective studies are warranted to validate these findings and explore additional predictive factors.

## Introduction

Intermittent exotropia (IXT) is one of the most common forms of strabismus [1], characterized by an outward deviation of the eyes that occurs intermittently. The etiology of IXT remains unclear, and its global prevalence ranges from 0.12 to 3.9% [1–4]. If left untreated, 15%−23% of cases worsen progressively [5,6]. Conservative treatments, such as part-time patching, orthoptic exercises, or botulinum toxin, are often the first treatment choice [7,8]. Surgical intervention is often required when conservative management fails to achieve satisfactory ocular alignment and control

**Data availability statement:** All relevant data are within the manuscript and its Supporting Information files.

**Funding:** The author(s) received no specific funding for this work.

**Competing interests:** The authors have declared that no competing interests exist.

progressively [5,6]. Various surgical procedures are performed, including unilateral or bilateral horizontal muscle surgery.

Recently, global reports on the success of surgical treatment for IXT have indicated a wide range of success rates, from 25.9% to 89%. The surgical outcomes varied depending on the types of procedure, including 70 to 86.2% for unilateral lateral recess [9–11], 46 to 64.5% for bilateral lateral rectus recession [9,12–15], 46 to 64.5% for unilateral lateral rectus recess-medial rectus resection [13,15,16], and 25.9 to 27.8% for unilateral lateral rectus recession-medial rectus plication [12,17].

Additionally, recent evidence has identified several factors influencing surgical success, such as postoperative consecutive esotropia (CE) [13,17–20], an age range of 3–5 years [21], a smaller preoperative angle [22], and the use of recess-resect procedures [15,23]. Despite these findings, significant variability remains in long-term surgical outcomes, particularly in the Thai population, where information is scarce. Studying this specific population may reveal unique factors or confirm previously identified ones, contributing valuable insights to the field.

This study aims to evaluate the success of surgical treatment for IXT and identify predictors of favorable outcomes in a tertiary referral center hospital.

## Materials and methods

This retrospective cohort study collected data from the outpatient and inpatient clinics of Phramongkutklao Hospital between January 1, 2013, and October 31, 2024. Researchers accessed data from February to April 2025. Data for individual participants was only accessed during data collection. The study was approved by the Institutional Review Board of Royal Thai Army Medical Department (approval number S090h/67_Exp). Due to its retrospective nature, the Institutional Review Board of Royal Thai Army Medical Department waived the need for informed consent. This study was conducted according to the principles of the Declaration of Helsinki.

### Study participants

This retrospective study involved 150 patients who underwent surgical correction for IXT at Phramongkutklao Hospital from January 2013 to October 2024. The inclusion criteria required a diagnosis of IXT as a basic type and a pseudo-divergence excess type, participants to have received two muscle surgeries, and a minimum follow-up of four visits over at least one year. Patients with preoperative angles greater than 50 prism diopters (PD), concurrent ocular or systemic conditions that could affect vision, and those with previous surgeries were excluded from the study.

### Disease definition

IXT is a type of strabismus characterized by intermittent outward deviation of one or both eyes, typically noticeable during periods of inattention or fatigue. Diagnosis is based on clinical examination, which includes assessing ocular alignment and measuring deviation angles using the prism and cover test.

## Surgical intervention

The surgical correction of IXT involves adjusting the extraocular muscles to address outward deviation. The two surgical techniques performed include:

- Bilateral lateral rectus recession, targeting both eyes for symmetric correction.
- Recess-resect procedures combine lateral rectus weakening with medial rectus strengthening in one eye.

For angles of 15 PD or greater, two horizontal muscle surgeries are performed. Traditionally, when a case exhibited a strongly dominant eye, unilateral recession-resection was the preferred method. If not, bilateral recession was employed.

The extent of muscle adjustment is calculated using the Marshall-Parks table [24], a standardized guideline for determining surgical dosages based on the preoperative deviation angle, as shown in Table 1. Three surgeons conducted the operations throughout the study period, encompassing all types of procedures.

## Data gathering

Preoperative data included age at the time of surgery, sex, angle of deviation, and sensory status. Visual acuity was assessed using the LogMAR scale, and stereopsis was evaluated with the Randot stereotest. The preoperative angle, used for surgical dosage, was the maximum angle previously measured, utilizing the larger angle from distant and near measurements. Subjective refraction was performed by a single optometrist to ensure consistency and was documented as a spherical equivalence. The surgical techniques employed included bilateral lateral rectus recession and unilateral lateral rectus recession combined with medial rectus resection.

Postoperative outcomes were assessed at multiple intervals: one day, one week, one month, three months, six months, and one year after surgery.

## Outcome definitions

Surgical success was defined as a postoperative deviation of exotropia of ≤10 prism PD and the absence of significant diplopia measured one year after surgery. Successful cases include early CE that later improves without diplopia or the need for reoperation. Early CE was defined as any esodeviation greater than 5 PD detected postoperatively on the alternate prism cover test. Diplopia was defined as a subjective symptom of seeing double vision, as reported by the patient or caregiver during clinical evaluation, without the use of formal prism or binocular vision testing. For stereopsis classification, "present" stereopsis referred to any measurable response on the Randot stereotest, whereas "absent" stereopsis indicated a complete lack of detectable stereopsis on this test.

Table 1. Surgical Dose Guidelines for Exotropia by Deviation Angle: Modified Marshall-Park Table [31].

| Angle of Deviation (prism diopters) | Bilateral surgery | | Unilateral surgery | |
|---|---|---|---|---|
| | Recess LR OU (mm) | Resect MR OU (mm) | Resect MR (mm) | Recess LR (mm) |
| 15 | 4 | 3 | 3 | 4 |
| 20 | 5 | 4 | 4 | 5 |
| 25 | 6 | 5 | 4.5 | 6 |
| 30 | 7 | 5.5 | 5 | 6.5 |
| 35 | 7.5 | 6 | 5.5 | 7 |
| 40 | 8 | 6.5 | 6 | 7.5 |
| 50 | 9 | NA | 6.5 | 8.5 |

Abbreviation: LR, lateral rectus; OU, both eyes; MR, medial rectus; mm, millimeter.

## Statistical analysis

Descriptive statistics were employed to summarize the study cohort's demographic characteristics, etiologies, clinical presentations, and treatment outcomes. Continuous variables were reported as means with standard deviations (SD), while categorical variables were presented as frequencies and percentages. Surgical outcomes were assessed based on postoperative ocular alignment measurements at various follow-up intervals.

Comparisons were conducted to evaluate differences between the motor success and non-success groups at one year postoperatively. Demographic variables (sex, age), baseline clinical parameters (visual acuity, stereopsis status and arcsec, refractive error, anisometropia, preoperative strabismic angle, onset duration, prior treatment), and surgical factors (procedure type, presence of CE at one week) were compared between the two groups. In addition, longitudinal comparisons of ocular alignment status (orthotropia, residual exotropia, and CE) were performed across postoperative time points (day 1, week 1, month 1, month 3, month 6, and year 1) to evaluate alignment trends over time.

The Mann-Whitney U test was used to analyze non-normally distributed continuous data from two groups for statistical comparison. Fisher's Exact test evaluated categorical variables in cases with small sample sizes or low expected frequencies. Logistic regression analysis was performed to determine independent predictors of surgical success; patients lacking stereopsis assessments were excluded from this analysis component.

A multivariable logistic regression model was constructed, incorporating adjustments for visual acuity, age, preexisting stereopsis status, the presence of CE on 1 week, and preoperative strabismic angle to identify independent factors influencing surgical outcomes. Variables were selected for inclusion in the model based on two criteria: statistical significance in univariable analyses ($p < 0.10$) and established clinical relevance reported in prior literature. Cases with missing preoperative stereopsis data (17 of 150 participants) were excluded from the logistic regression analysis using a complete-case approach. This method was selected to preserve the validity of regression estimates and avoid potential bias associated with imputing stereopsis values, given the clinical nature of the variable and its role as a predictor in the analysis.

Model performance was evaluated using several diagnostic measures. Calibration was assessed with the Hosmer–Lemeshow goodness-of-fit test, while discrimination was examined using the area under the receiver operating characteristic (ROC) curve. Multicollinearity among predictor variables was evaluated using variance inflation factors (VIFs), with values <2 considered acceptable.

Statistical significance was set at a p-value of < 0.05. All analyses used Stata version 14 (StataCorp LLC, USA).

## Results

During the study period, there were 322 participants with exotropia. Some cases were excluded due to incomplete medical records, previous surgeries, insufficient follow-up, or the presence of intraocular and intracranial abnormalities. The recruitment of participants is illustrated in Fig 1. The study cohort comprised 150 participants, with a mean age of 15.3 ± 12.2 years (ranging from 1 to 66 years). The mean follow-up period was 14.25 ± 2.35 months. The median follow-up was 14.3 months (interquartile range [IQR], 12.6–15.8 months). The average visual acuity was 0.19 ± 0.21 LogMar, and the mean refraction was −1.12 ± 2.81 diopters. Stereopsis evaluation records were available for 88.67% (133/150) of participants, with 83.46% (111/133) showing detection of stereopsis. The mean preoperative stereopsis was measured at 135.5 ± 189.7 arcseconds. The average onset duration was 5.08 ± 5.39 years, and the mean preoperative deviation was 37.8 ± 7.2 PD. Demographic data can be found in Table 2.

Bilateral muscle surgery was the most common procedure in this study, with bilateral lateral rectus recession performed in 56% (84/150) of cases and unilateral medial rectus resection combined with lateral rectus recession in 44% (66/150). No statistically significant differences were observed in surgical outcomes among the various procedures.

Postoperative data are presented in Table 3. The mean postoperative angle was 9.5 ± 7.4 in one year. At the one-year postoperative follow-up, 111 patients (74%) achieved surgical success. The initial six months of postoperative follow-up

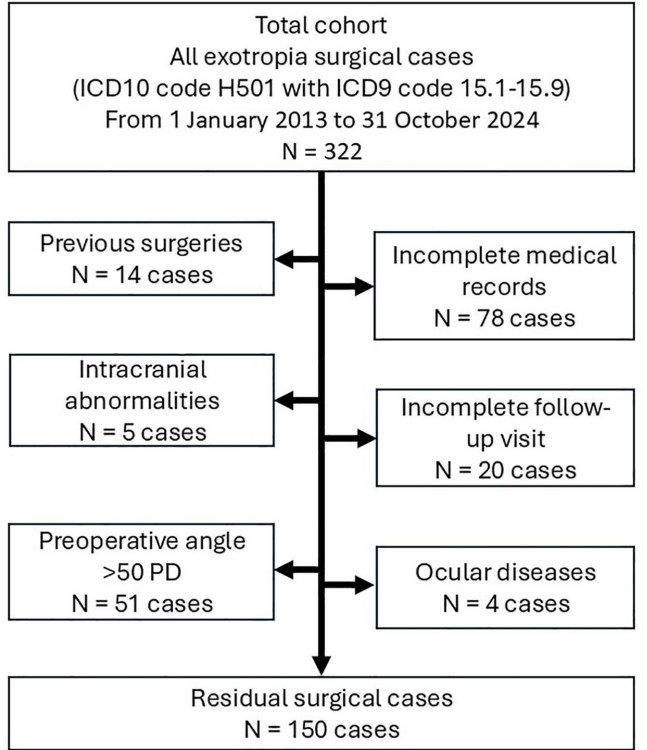

**Fig 1. Flowchart of participants' recruitment.** Abbreviations: ICD, International Classification of Diseases; N, number of participants.

were significantly associated with successful outcomes. The status of postoperative stereopsis improved to 83.46% (111/133) one year after surgery. The mean postoperative stereopsis was 120.6 ± 188.9.

Postoperative alignment patterns demonstrated distinct trajectories between success and non-success groups over the one-year follow-up period, as illustrated in **Fig 2**. In the success group, CE was most frequent on day 1 (45 cases; 30.0%) but rapidly declined to nearly absent by three months (1 case; 0.7%) and remained minimal thereafter. Orthotropia was highest in the early postoperative period (41–42 cases; 27.3–28.0% at day 1 and week 1) but progressively decreased over time, while residual exotropia gradually increased, becoming the predominant alignment by one year (105 cases; 70.0%). In contrast, the non-success group was consistently characterized by high rates of residual exotropia across all time points (26 cases; 17.3% on day 1, increasing to 38 cases; 25.3% at one year), with orthotropia nearly absent after week 1 (≤2 case; ≤ 1.3%) and only minimal CE observed throughout follow-up (≤5 cases; ≤ 3.3%).

Among cases with measurable stereopsis, the Chi-square test indicated significantly better surgical outcomes in patients with some preoperative stereopsis compared to those with absent stereopsis. One case showed improvement from absent stereopsis, while five cases exhibited enhanced stereopsis. Fig 3 illustrates a comparison of preoperative and postoperative stereopsis in successful versus non-successful surgical outcome groups. In contrast, no statistically significant differences were observed in stereopsis levels between the successful and non-successful surgical outcome groups when comparing preoperative and postoperative measurements.

Table 4 shows logistic regression analysis. Multivariable analysis identified preoperative detection of stereopsis and the occurrence of postoperative CE at 1 week as significant predictors of successful surgical outcomes, with adjusted odds ratios of 2.81 (95% CI, 1.14–6.93) and 4.56 (95% CI, 1.60–12.96), respectively.

**Table 2. Analysis of Patient Demographics and Preoperative Clinical Characteristics Using Mann-Whitney U and Fisher's Exact Tests.**

| Variable | Success | Non-success | p-value |
|---|---|---|---|
| Age | | | |
| Mean ± SD | 17.06 ± 16.09 | 12.64 ± 11.08 | 0.092 |
| Median (Min-Max) | 10.0 (1-66) | 9.0 (2-53) | |
| Sex | | | 0.093 |
| Male | 52 (46.8%) | 25 (64.1%) | |
| Female | 59 (53.2%) | 14 (35.9%) | |
| Visual Acuity (LogMAR) | | | 0.206 |
| Mean ± SD | 0.17 ± 0.23 | 0.20 ± 0.19 | |
| Median (Min-Max) | 0.1 (0-2) | 0.1 (0-0.8) | |
| Stereopsis | | | 0.006 |
| Present | 89 (89%) | 22 (66.7%) | |
| Absent | 11 (11%) | 11 (33.3%) | |
| Preoperative Stereopsis (arcsec) | | | 0.520 |
| Mean ± SD | 115.9 ± 186.5 | 104.5 ± 162.7 | |
| Median (Min-Max) | 60 (0–1400) | 40 (0–800) | |
| Refraction (D) | | | 0.285 |
| Mean ± SD | −1.15 ± 2.20 | −0.91 ± 2.77 | |
| Median (Min-Max) | −0.5 (−8.0 to 4.25) | −0.5 (−11.0 to 5.0) | |
| Anisometropia (D) | | | 0.799 |
| Mean ± SD | 0.56 ± 0.95 | 0.72 ± 1.83 | |
| Median (Min-Max) | 0.25 (0-6) | 0.25 (0-11) | |
| Duration (Year) | | | 0.913 |
| Mean ± SD | 4.64 ± 5.01 | 4.85 ± 4.34 | |
| Median (Min-Max) | 3 (0.5-30) | 3 (0.5-20) | |
| Prior Treatment | | | 0.761 |
| Glasses | 21 (18.9%) | 9 (23.1%) | |
| Orthoptic | 1 (0.9%) | 1 (2.6%) | |
| Patching | 1 (0.9%) | 2 (5.1%) | |
| Glasses + Orthoptic | 4 (3.6%) | 0 (0%) | |
| Orthoptic + Patching | 24 (21.6%) | 8 (20.5%) | |
| Glasses + Orthoptic + Patching | 33 (29.7%) | 8 (20.5%) | |
| No Treatment | 27 (24.3%) | 11 (28.2%) | |
| Type of Surgical Procedure | | | 0.136 |
| Recess-resect | 53 (47.7%) | 13 (33.3%) | |
| Bilateral recess | 58 (52.3%) | 26 (66.7%) | |
| Preoperative Angle (Prism Diopters) | | | |
| Mean ± SD | 36.86 ± 8.99 | 39.23 ± 9.70 | 0.136 |
| Median (Min-Max) | 40 (18-50) | 40 (20-50) | |

**Notes:**

- p-value ≤ 0.05 indicates statistical significance

- All continuous variables were analyzed using the Mann-Whitney U test.

- Categorical variables were analyzed using Fisher's Exact test.

- Motor success: Postoperative exotropia ≤10 prism diopters (PD) without significant diplopia at 1 year.

- Sensory success: "Present" stereopsis = any measurable response on the Randot stereotest; "Absent" = no detectable stereopsis.

**Table 3. Postoperative Surgical Outcomes Analyzed with Mann-Whitney U and Fisher's Exact Tests.**

| Status/ Angle | Day 1 | 1 Week | 1 Month | 3 Months | 6 Months | 1 year |
|---|---|---|---|---|---|---|
| **Orthotropia** | | | | | | |
| - Success | 41 (27.3%) | 42 (28.0%) | 42 (28.0%) | 31 (20.7%) | 16 (10.7%) | 5 (3.3%) |
| - Non-success | 8 (5.3%) | 2 (1.3%) | 0 (0.0%) | 0 (0.0%) | 0 (0.0%) | 0 (0.0%) |
| **Residual Exotropia** | | | | | | |
| - Success | 25 (16.7%) | 46 (30.7%) | 58 (38.7%) | 79 (52.7%) | 94 (62.7%) | 105 (70%) |
| - Non-success | 26 (17.3%) | 34 (22.7%) | 37 (24.7%) | 37 (24.7%) | 38 (25.3%) | 38 (25.3%) |
| **Consecutive Esotropia** | | | | | | |
| - Success | 45 (30.0%) | 23 (15.3%) | 11 (7.3%) | 1 (0.7%) | 1 (0.7%) | 1 (0.7%) |
| - Non-success | 5 (3.3%) | 3 (2.0%) | 2 (1.3%) | 2 (1.3%) | 1 (0.7%) | 1 (0.7%) |
| **Angle (PD)** | | | | | | |
| Mean ± SD | 6.7 ± 6.9 | 6.7 ± 6.5 | 6.6 ± 6.6 | 7.0 ± 6.5 | 8.1 ± 7.0 | 9.5 ± 7.4 |
| Median (Min–Max) | 4.5 (0–35) | 6.0 (0–30) | 4.0 (0–30) | 6.0 (0–25) | 6.0 (0–35) | 8.0 (0–36) |

Notes:

- Data are presented as n (%) for categorical outcomes and as mean ± SD or median (min–max) for continuous data.

- All angle measurements are reported as absolute deviation magnitudes in positive prism diopters (PD), regardless of direction (i.e., both exodeviation and esodeviation are included as positive values).

- Motor success: Postoperative exotropia ≤10 prism diopters (PD) without significant diplopia at 1 year.

- Sensory success: "Present" stereopsis = any measurable response on the Randot stereotest; "Absent" = no detectable stereopsis.

- Consecutive esotropia group includes patients with esodeviation >5 PD postoperatively.

- Residual exotropia includes postoperative exodeviation >10 PD.

- Percentages are calculated within each outcome group (Success/Non-success).

- p-values are from Fisher's Exact Test (status) and Mann-Whitney U Test (angle).

Abbreviation: PD = prism diopter; SD = standard deviation.

The multivariable logistic regression model demonstrated good calibration by the Hosmer–Lemeshow test (p = 0.41) and acceptable discrimination, with an area under the ROC curve of 0.77. Assessment of multicollinearity indicated no significant concerns, as all VIFs were <2.

A visual summary of surgical outcomes stratified by preoperative stereopsis and the presence of CE is presented in Fig 4. Patients with detectable preoperative stereopsis demonstrated higher success rates (82.5%) compared with those lacking stereopsis (66.7%). Similarly, cases that developed CE postoperatively achieved the highest success rate (88.5%) relative to those without CE (73.9%). These findings further underscore the prognostic significance of both sensory status and postoperative alignment patterns in predicting long-term surgical outcomes.

The average dose-response for each surgical procedure was calculated based on the postoperative ocular alignment at one year. In patients who underwent the unilateral recess-resect procedure, the average dose-response was −3.97 ± 1.07 PD/mm for lateral rectus (LR) and −6.21 ± 1.57 PD/mm for medial rectus (MR). In patients who underwent bilateral lateral rectus recession, the average dose-response was −3.15 ± 1.09 PD/mm. Among the surgical techniques, the unilateral recess-resect procedure demonstrated a more significant dose-response compared to the bilateral lateral rectus recession technique.

## Discussion

Our study found a motor success rate of 74%, consistent with related research reporting success rates ranging from 25.9% to 89% [9,10,12–18,21–23,25–37]. Additionally, our sensory outcomes were stereopsis detection in 83.46% of cases, which is slightly better than the related reported sensory success rates of 42.1% to 78% [16,28,38]. Limited data

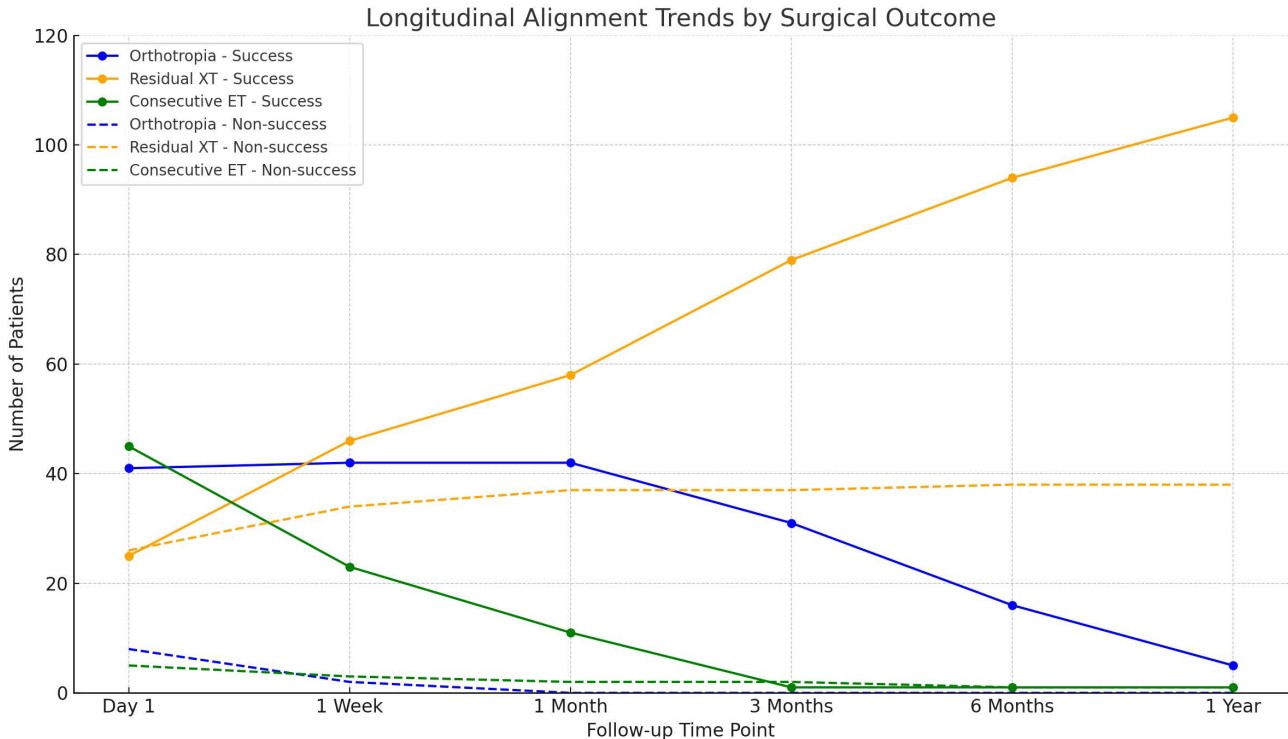

**Fig 2. Longitudinal Alignment Trends After Surgery.** Abbreviations: XT, exotropia;ET, esotropia.

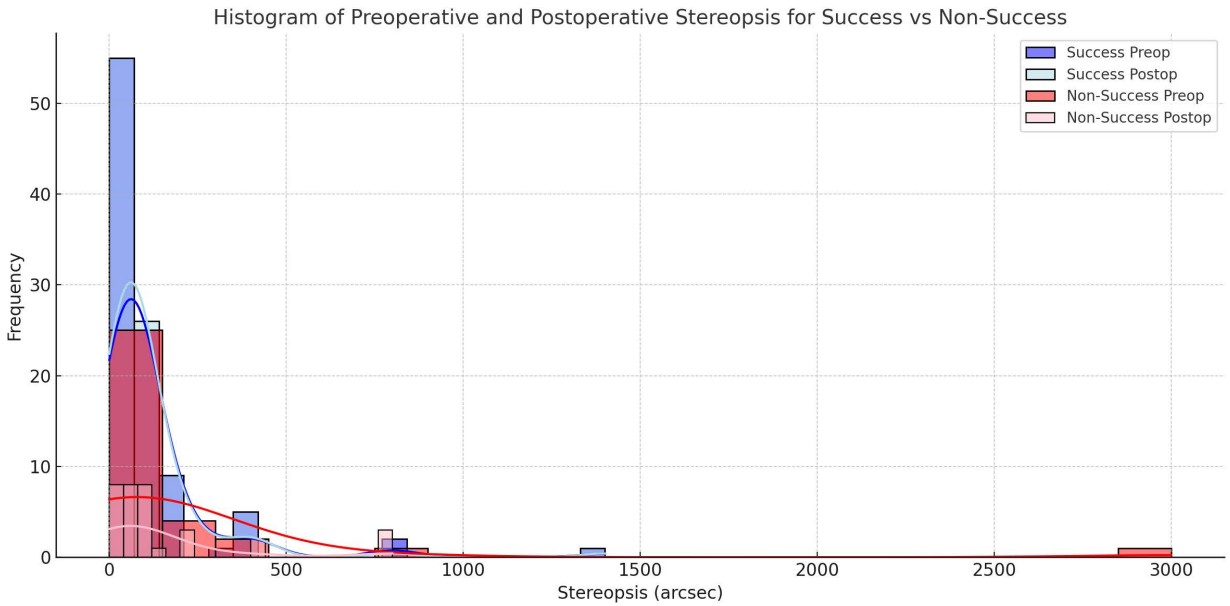

**Fig 3. Comparison of Preoperative and Postoperative Stereopsis in Success vs Non-Success Surgical Outcome Groups.**

**Table 4. Multivariable Analysis of Factors Influencing Surgical Success in Intermittent Exotropia Surgery.**

| Variable | p-value | Adjusted OR | 95% CI (Lower – Upper) |
|---|---|---|---|
| Overcorrection at 1 week | 0.004 | 4.56 | 1.60–12.96 |
| Preoperative stereopsis | 0.025 | 2.81 | 1.14–6.93 |
| Preoperative strabismus angle (PD) | 0.608 | 0.99 | 0.94–1.03 |
| Preoperative visual acuity (LogMAR) | 0.581 | 1.64 | 0.29–9.37 |
| Age (years) | 0.171 | 1.02 | 0.99–1.06 |

Note: - Percentages for continuous variables (preop angle, visual acuity) are not shown because they are not categorical.

- Motor success: Postoperative exotropia ≤10 prism diopters (PD) without significant diplopia at 1 year.

- Sensory success: "Present" stereopsis = any measurable response on the Randot stereotest; "Absent" = no detectable stereopsis.

Abbreviations: OR = odds ratio; CI = confidence interval; PD = prism diopters.

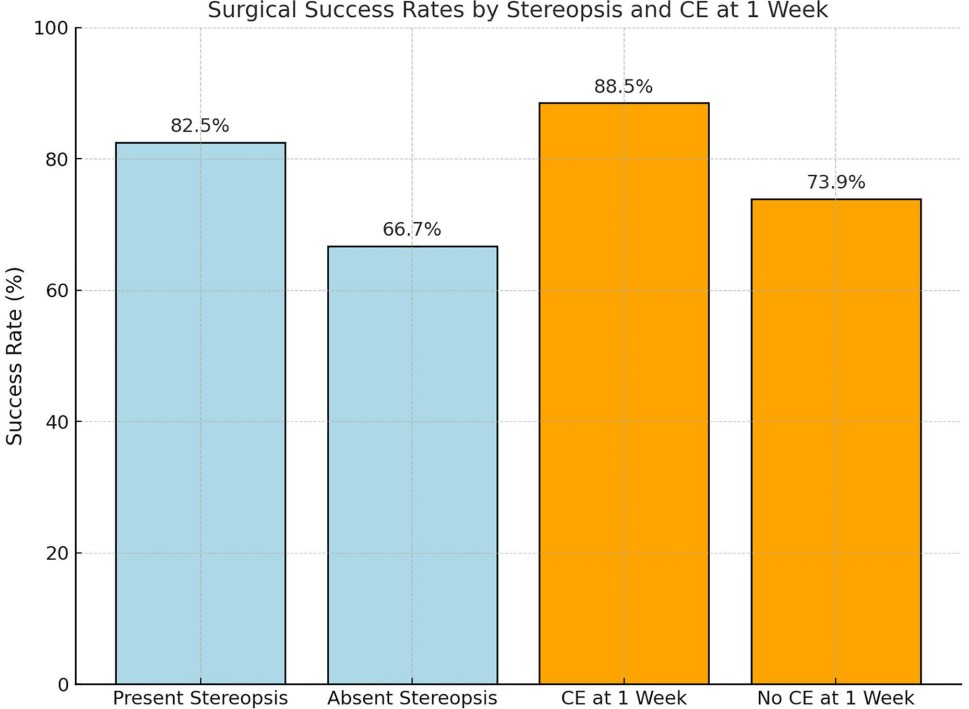

**Fig 4. Surgical success rates stratified by preoperative stereopsis and consecutive esotropia.** Abbreviations: CE, consecutive esotropia.

exist concerning surgical success rates for IXT in the Thai population; however, existing reports indicate motor success rates between 69% and 84.19% [35,39,40] and stereopsis success rates between 32.48% and 43% [35,39].

Consistent with previous studies, our findings (Fig 4) suggest that the presence of preoperative stereopsis and the development of CE are strong prognostic indicators of favorable motor outcomes following IXT surgery. Preoperative stereopsis and early CE emerged as key factors influencing surgical success in this study. While the relationship between preoperative stereopsis and surgical outcomes in IXT surgery remains inconclusive, previous studies provide valuable insights. Beneish et al. [36] reported a 60% motor success rate at an average follow-up of 3.3 years among 67 participants with IXT, highlighting that poor preoperative stereopsis did not adversely affect surgical outcomes and was associated with favorable long-term results, particularly when combined with early postoperative CE. Similarly, Lee et al. [37]

found no significant association between preoperative stereopsis and surgical success in a retrospective review of 137 participants. Their study noted that patients with better stereopsis were generally older at the time of IXT detection and exhibited superior best-corrected visual acuity compared to those with poorer stereopsis. Yildirim et al. [41] conducted a prospective study of 26 cases and 112 controls, achieving a 69% success rate in ocular alignment, though they observed a trend toward reduced surgical success in patients with central fusion. In alignment with our findings, this study demonstrated that 58% of participants experienced improved distant stereopsis post-surgery, with successful cases achieving better stereopsis than unsuccessful ones. While preoperative stereopsis may not consistently predict surgical success, our results suggest that patients with some degree of stereopsis are more likely to achieve favorable surgical outcomes and enhanced stereopsis after surgery, underscoring the importance of preoperative stereopsis evaluation in surgical planning for IXT.

Our study identifies early CE following surgery for IXT as a significant predictive factor for surgical success, consistent with previous findings suggesting that mild overcorrection immediately after surgery is associated with improved long-term outcomes [13,17,18,33]. Early postoperative CE, characterized by esodeviation during the initial follow-up period, may indicate effective surgical alignment and a reduced risk of recurrence. This phenomenon likely stems from a balance between motor alignment and sensory adaptation, where CE enhances fusional adaptation, promotes stable motor control, and encourages the restoration of sensory fusion, all of which are critical for maintaining long-term alignment. While the optimal range of overcorrection remains debated, with prior studies suggesting an ideal range of 5–16 PD [13,17,18,33], excessive overcorrection risks diplopia and disruption of binocular function, whereas undercorrection increases the likelihood of recurrent exotropia. In our study, patients with early CE exhibited successful outcomes without significant complications, supporting the benefit and tolerability of mild overcorrection. Notably, immediate postoperative deviation measured on the day of surgery or postoperative day 1 was often unreliable due to patient discomfort or limited cooperation; therefore, I evaluated initial postoperative deviation one week after surgery. During this interval, most patients experienced exodrift, leading to a lower proportion of overcorrected patients in our study compared to others.

In our cohort, longitudinal alignment trends revealed a rapid decline in CE after the immediate postoperative period, whereas residual exotropia progressively increased and became the predominant alignment pattern at one year (Fig 2). This trajectory differs from prior studies in which early CE is often considered a favorable prognostic indicator for long-term alignment stability in IXT [13,17–20,33]. In our data, however, patients exhibiting residual exotropia at early follow-up were more likely to achieve long-term motor success compared with those who initially demonstrated CE. This apparent discrepancy may stem from differences in the degree and duration of CE observed in our cohort; most cases of CE in this study were mild and resolved spontaneously within three months, while persistent or marked CE was rare. Conversely, residual exotropia was common but often within a small deviation range, which may still fall within our success criteria (≤10 PD) and thus be classified as successful. Additionally, variations in surgical dose, patient age, and stereopsis status might have contributed to this divergence from previously reported trends. These findings highlight the need for cautious interpretation of early postoperative alignment, emphasizing that both CE and residual exotropia trajectories should be evaluated in the context of their magnitude and temporal resolution rather than their presence alone.

Our postoperative data revealed that initial motor success was significantly high within the first six months following surgery. This early success is likely due to the immediate effectiveness of the surgical correction in realigning the eyes. However, the success rate declined during subsequent follow-up periods. This decline may result from the gradual weakening of surgical effects, natural tendencies of ocular muscles to revert, and potential challenges in maintaining alignment over time. These findings underscore the importance of long-term follow-up and ongoing interventions to sustain surgical outcomes.

The findings of this study indicate important clinical implications for managing IXT. Preoperative stereopsis has been shown to significantly influence successful surgical outcomes, underscoring the importance of a thorough preoperative assessment of stereopsis status. Patients with preoperative stereopsis tend to have better surgical success rates and

exhibit potential gains in stereoacuity postoperatively. Moreover, postoperative CE in the first week has been shown to influence successful surgical outcomes in this context. Additionally, the initial ocular alignment achieved within the first six months after surgery is crucial, as it tends to be sustained over time; however, a gradual decline in alignment may occur with longer follow-up. These insights highlight the necessity of regular and long-term postoperative monitoring to maintain and optimize surgical outcomes.

## Conclusion

In this study, motor outcomes showed good surgical success at one-year follow-up; moreover, favorable sensory outcomes were consistently achieved. There was no significant difference among the various surgical techniques used. However, pre-operative detection of stereopsis and the presence of postoperative consecutive esotropia were significantly associated with successful surgical outcomes for IXT. This study provides valuable insights into the factors that influence successful outcomes, assisting clinicians in optimizing treatment strategies for improved patient prognosis.

### Study limitations and future directions

This study has several limitations. Its retrospective design introduces potential selection bias and limits control over confounding variables. The single-center setting and variability in surgical techniques, including surgeon expertise, may affect generalizability. Moreover, inter-surgeon variability was not analyzed in detail, and differences in surgical technique or experience among the three operating surgeons may have influenced postoperative outcomes. Additionally, the mean one-year follow-up period may be insufficient to assess long-term stability. Our binocular vision assessment relied solely on the Randot stereotest for near vision, which may underestimate full binocular function by omitting distant stereopsis evaluation. The small sample size was affected by loss to follow-up, which was necessary to meet the inclusion criteria, and this may have influenced the statistical power of this study. The control score was assessed during clinical follow-up and influenced surgical decision-making; however, it was not included in the final analysis due to inconsistent documentation in the retrospective dataset. Additionally, neurologic status was not formally evaluated or recorded in this study, which may have limited the assessment of potential confounding factors. The retrospective nature of the dataset limited consistent documentation of recurrence or failure timing, precluding time-to-event analysis to assess long-term surgical durability. Finally, we excluded deviations greater than 50 PD, as these exceed the Marshall-Parks table's recommended two-muscle correction range—a decision that prioritizes internal validity but limits applicability to large-angle exotropia.

Prospective multicenter studies with standardized surgical protocols and extended follow-up are essential to validate these findings and assess long-term outcomes. Comparative studies should examine augmented approaches (e.g., three-muscle surgery or adjustable sutures) for large-angle deviations, while incorporating both near and distant stereopsis assessments to fully characterize binocular function.

## Supporting information

**S1. Completed STROBE Checklist.**
(DOCX)

**S2. IXT all case for calculate 2025 21-8-25.**
(XLSX)

## Author contributions

**Conceptualization:** Worapot Srimanan, Phawasutthi Keokajee.

**Data curation:** Worapot Srimanan, Phawasutthi Keokajee.

**Formal analysis:** Worapot Srimanan, Phawasutthi Keokajee.

**Funding acquisition:** Worapot Srimanan.

**Investigation:** Worapot Srimanan, Phawasutthi Keokajee.

**Methodology:** Worapot Srimanan.

**Project administration:** Worapot Srimanan.

**Resources:** Worapot Srimanan.

**Supervision:** Worapot Srimanan, Phawasutthi Keokajee.

**Validation:** Worapot Srimanan, Phawasutthi Keokajee.

**Visualization:** Worapot Srimanan, Phawasutthi Keokajee.

**Writing – original draft:** Worapot Srimanan, Phawasutthi Keokajee.

**Writing – review & editing:** Worapot Srimanan.

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
