## [Decision Letter · Decision Letter 0]

31 Jul 2025

Dear Dr. Srimanan,

Thank you for submitting your manuscript to PLOS ONE. After careful consideration, we feel that it has merit but does not fully meet PLOS ONE’s publication criteria as it currently stands. Therefore, we invite you to submit a revised version of the manuscript that addresses the points raised during the review process.

**Please look into the comments of all the reviewers.**

We look forward to receiving your revised manuscript.

Kind regards,

PremNandhini Satgunam

Academic Editor

PLOS ONE

Journal Requirements:

3. Please include captions for your Supporting Information files at the end of your manuscript, and update any in-text citations to match accordingly. Please see our Supporting Information guidelines for more information: http://journals.plos.org/plosone/s/supporting-information .

Additional Editor Comments:

The reviewers have given a careful consideration and have identified areas that needs better reporting, clarity for terms used and explanation for statistical methods. I hope the authors can take these comments and work on their manuscript to improve the quality of this work.

Reviewers' comments:

Reviewer's Responses to Questions

**Comments to the Author**

1. Is the manuscript technically sound, and do the data support the conclusions?

Reviewer #1: No

Reviewer #2: Partly

Reviewer #3: Yes

2. Has the statistical analysis been performed appropriately and rigorously?

Reviewer #1: Yes

Reviewer #2: No

Reviewer #3: Yes

3. Have the authors made all data underlying the findings in their manuscript fully available?

Reviewer #1: No

Reviewer #2: Yes

Reviewer #3: Yes

4. Is the manuscript presented in an intelligible fashion and written in standard English?

Reviewer #1: Yes

Reviewer #2: Yes

Reviewer #3: Yes

Reviewer #1: Clarification of Table 3 Interpretation:

Table 3 reports postoperative alignment status at various time points grouped by 1-year surgical outcomes. However, the finding that patients with residual exotropia had higher success rates than those with early consecutive esotropia (CE) appears inconsistent with commonly observed postoperative trajectories in IXT, where early CE is typically a favorable prognostic factor. This discrepancy may confuse readers. The authors are advised to clarify this point in the Results or Discussion section and consider including a line graph showing longitudinal alignment trends (CE, orthotropia, exotropia) over time in both success and non-success groups.

Definition of Sensory Success and Stereopsis Threshold:

The term “sensory outcomes were successful in 85.5% of cases” lacks a precise definition of "success." It appears the presence of any measurable stereopsis was regarded as success, but this is neither explicitly defined nor supported by comparison to existing literature. Prior studies often use more specific criteria (e.g., stereopsis ≤100 arcsec) for sensory success. The authors should clearly define the threshold used, justify it in the context of the literature, and acknowledge any limitations if a less stringent criterion was applied.

Definition of “Absent” Stereopsis in Table 2:

Table 2 refers to stereopsis as "present" or "absent," but the manuscript does not define what constitutes “absent” stereopsis. The authors should specify whether this refers to a complete lack of measurable stereopsis on the Randot test, and clarify the cutoff or conditions under which stereopsis was considered absent.

Reviewer #2: Dear Authors,

Thank you for the opportunity to review this interesting and clinically relevant manuscript. The study investigates surgical outcomes in intermittent exotropia (IXT) over a 9-year period, which is important for both regional and global understanding of treatment success and predictive factors. The manuscript is well-structured, and the research question is clearly defined. However, several methodological and reporting aspects need clarification and improvement to enhance the validity and reproducibility of your findings.

A. Major Concerns

1. Definition of Postoperative Consecutive Esotropia (CE): Since early CE was identified as a significant predictor of surgical success, please clarify the clinical cutoff used to define CE (e.g., >5 PD or >10 PD). A precise definition is essential for reproducibility and for clinicians to apply these findings in practice.

2. Numerical Inconsistency in Table 3: In Table 3 (Day 1), the total number of patients across alignment categories does not equal 150. Please verify whether this is due to missing follow-up data or a reporting error, and provide an explanation or corrected table as appropriate.

3. Sample Size vs Study Duration:Over a 9-year period, a sample size of 150 appears relatively small for a tertiary center. Please clarify whether this was due to strict eligibility criteria, loss to follow-up, or exclusion based on comorbidities or incomplete records. Discuss any resulting implications for generalizability and statistical power in your limitations.

4. Handling of Missing Stereopsis Data: Only 131 of 150 participants had preoperative stereopsis data. Please clarify how missing data were handled in the logistic regression analysis (e.g., complete-case analysis, imputation, or exclusion).

5. Statistical Model Reporting: The multivariate logistic regression is well-conceived, but the analysis lacks model validation metrics. Consider adding or commenting on:

a. Goodness-of-fit test (e.g., Hosmer-Lemeshow).

b. Discrimination (e.g., AUC/ROC curve).

3. Multicollinearity checks.

These would strengthen confidence in the identified predictors.

6. Definition of Success in Tables: Please reiterate your criteria for motor and sensory success in the footnotes of Tables 2–4. This will improve clarity and consistency for the reader.

7. Surgeon Variability: Since three surgeons performed procedures over the study period, please clarify whether inter-surgeon variability was assessed or considered as a confounding factor.

8. references : most of the references are before 2020, I would request authors to look into new research articles and comparison with recent literature.

B. Minor Suggestions

1. Please ensure Figures 1 and 2 are included and fully labeled in the revised manuscript. Their absence currently limits interpretation of key findings.

2. You may consider including a visual summary (e.g., boxplot or bar graph) of surgical outcomes stratified by presence/absence of stereopsis and CE.

3. If feasible in future studies, include distant stereopsis testing, as IXT control is often more relevant at distance fixation.

Finally in conclusion,this study provides valuable insight into IXT management and postoperative outcomes in a population that is underrepresented in the literature. I would like encourage revision to clarify definitions, correct inconsistencies, and improve the statistical reporting. With these revisions, the manuscript will offer meaningful contributions to the field of pediatric and strabismus surgery.

Best regards

Reviewer #3: As the statistical reviewer I will focus on methods and reporting.

Major

1) the sample is small and power will be limited, especially in a multivariable regression framework. findings need to be discussed carefully and not as definitive.

2) the methods section needs to be written with more clarity about the comparisons made, e.g. in what variables and for what purposed.

3) the STROBE statement has been submitted but I cannot find information as to how missing data were handled. Considering levels of missingness, why weren't multiple imputation approaches considered?

4) Near stereopsis outcome has 3 categories and it was not clear if and how analysed.

Minor

1) multivariable not multivariate (multiple outcomes) regression.

2) how were variables selected for inclusion in the multivariable logistic regression

**Do you want your identity to be public for this peer review?** For information about this choice, including consent withdrawal, please see our Privacy Policy

Reviewer #1: **Yes: ** HYUN JIN SHIN

Reviewer #2: No

Reviewer #3: No

---

## [Author Response · Author response to Decision Letter 1]

23 Aug 2025

Response Letter PLOS One

PONE-D-25-34023

Early Postoperative Alignment and Preoperative Stereopsis as Determinants of Success in Intermittent Exotropia Surgery

PLOS ONE

Dear Dr. Srimanan,

Thank you for submitting your manuscript to PLOS ONE. After careful consideration, we feel that it has merit but does not fully meet PLOS ONE’s publication criteria as it currently stands. Therefore, we invite you to submit a revised version of the manuscript that addresses the points raised during the review process.

Please look into the comments of all the reviewers.

We look forward to receiving your revised manuscript.

Kind regards,

PremNandhini Satgunam

Academic Editor

PLOS ONE

Journal Requirements:

Additional Editor Comments:

The reviewers have given a careful consideration and have identified areas that needs better reporting, clarity for terms used and explanation for statistical methods. I hope the authors can take these comments and work on their manuscript to improve the quality of this work.

Response: Thank you for reviewing and providing suggestions for our manuscript. We thoroughly reviewed and revised the manuscript based on your expert recommendations. We hope our revised manuscript meets your high standards and has the potential to be published in your esteemed journal.

Reviewers' comments:

Reviewer's Responses to Questions

Comments to the Author

1. Is the manuscript technically sound, and do the data support the conclusions?

Reviewer #1: No

Reviewer #2: Partly

Reviewer #3: Yes

2. Has the statistical analysis been performed appropriately and rigorously?

Reviewer #1: Yes

Reviewer #2: No

Reviewer #3: Yes

3. Have the authors made all data underlying the findings in their manuscript fully available?

Reviewer #1: No

Reviewer #2: Yes

Reviewer #3: Yes

4. Is the manuscript presented in an intelligible fashion and written in standard English?

Reviewer #1: Yes

Reviewer #2: Yes

Reviewer #3: Yes

5. Review Comments to the Author

Reviewer #1: Clarification of Table 3 Interpretation:

Table 3 reports postoperative alignment status at various time points grouped by 1-year surgical outcomes. However, the finding that patients with residual exotropia had higher success rates than those with early consecutive esotropia (CE) appears inconsistent with commonly observed postoperative trajectories in IXT, where early CE is typically a favorable prognostic factor. This discrepancy may confuse readers. The authors are advised to clarify this point in the Results or Discussion section and consider including a line graph showing longitudinal alignment trends (CE, orthotropia, exotropia) over time in both success and non-success groups.

Response: Thank you for your valuable suggestion. The description of the surgical success at one year postoperative was thoroughly detailed in the results and discussion sections of the manuscript, specifically on page 11, lines 211-220, and pages 14-15, lines 310-323. The longitudinal alignment trends graph is also included in Figure 2.

Definition of Sensory Success and Stereopsis Threshold:

The term “sensory outcomes were successful in 85.5% of cases” lacks a precise definition of "success." It appears the presence of any measurable stereopsis was regarded as success, but this is neither explicitly defined nor supported by comparison to existing literature. Prior studies often use more specific criteria (e.g., stereopsis ≤100 arcsec) for sensory success. The authors should clearly define the threshold used, justify it in the context of the literature, and acknowledge any limitations if a less stringent criterion was applied.

Response: Thank you for your valuable suggestion. We apologize for the incorrect mention of the success of sensory outcomes. The sensory status only referred to the detection of stereopsis, which was identified by the near stereoacuity test, but it was not a successful sensory outcome. Thank you for your insightful comment. The sentence was rewritten for greater clarity and a more academic tone, as suggested on pages 5-6, lines 122-124. Furthermore, the success of sensory status in the manuscript was clarified through rewriting, and the term “stereopsis detection” was replaced accordingly on page 13, line 271.

Definition of “Absent” Stereopsis in Table 2:

Table 2 refers to stereopsis as "present" or "absent," but the manuscript does not define what constitutes “absent” stereopsis. The authors should specify whether this refers to a complete lack of measurable stereopsis on the Randot test, and clarify the cutoff or conditions under which stereopsis was considered absent.

Response: Thank you for your valuable suggestion. The sensory status was only the detection of any stereopsis, not a confirmation of successful sensory function. Thank you for your excellent comment. The sentence was revised for greater clarity and a more academic tone, as suggested, in the method section of the revised manuscript, pages 5-6, lines 122-124.

Reviewer #2: Dear Authors,

Thank you for the opportunity to review this interesting and clinically relevant manuscript. The study investigates surgical outcomes in intermittent exotropia (IXT) over a 9-year period, which is important for both regional and global understanding of treatment success and predictive factors. The manuscript is well-structured, and the research question is clearly defined. However, several methodological and reporting aspects need clarification and improvement to enhance the validity and reproducibility of your findings.

A. Major Concerns

1. Definition of Postoperative Consecutive Esotropia (CE): Since early CE was identified as a significant predictor of surgical success, please clarify the clinical cutoff used to define CE (e.g., >5 PD or >10 PD). A precise definition is essential for reproducibility and for clinicians to apply these findings in practice.

Response: Thank you for your valuable suggestion. The exact definition of surgical success has been written. The cut-off value for early consecutive esotropia > 5 prism diopters was noted in the method section of the revised manuscript on page 5, lines 121-122.

2. Numerical Inconsistency in Table 3: In Table 3 (Day 1), the total number of patients across alignment categories does not equal 150. Please verify whether this is due to missing follow-up data or a reporting error, and provide an explanation or corrected table as appropriate.

Response: Thank you for your valuable suggestion. We apologize for an error in data presentation. Thank you for your careful evaluation. The reporting error was corrected to reflect exactly 150 total cases at each point. Table 3 was rechecked, and the correct numbers were inserted into each cell.

3. Sample Size vs Study Duration:Over a 9-year period, a sample size of 150 appears relatively small for a tertiary center. Please clarify whether this was due to strict eligibility criteria, loss to follow-up, or exclusion based on comorbidities or incomplete records. Discuss any resulting implications for generalizability and statistical power in your limitations.

Response: Thank you for your valuable suggestion. The small sample size was due to the study's strict inclusion criteria. Most patient losses to follow-up aligned with the inclusion criteria, resulting in a small number of participants. The discussion was mentioned in the limitations part of the manuscript on page 16, lines 356-357.

4. Handling of Missing Stereopsis Data: Only 131 of 150 participants had preoperative stereopsis data. Please clarify how missing data were handled in the logistic regression analysis (e.g., complete-case analysis, imputation, or exclusion).

Response: Thank you for this valuable comment. We acknowledge that preoperative stereopsis data were missing for 19 participants. For the logistic regression analysis, we performed a complete-case analysis, excluding cases with missing stereopsis data from the model. This approach was chosen to avoid potential bias from imputation and to ensure the robustness of the regression estimates. The final analysis included only participants with complete stereopsis data. The handling of missing stereopsis data excluded from logistic regression was added in the methods section of the revised manuscript on page 7, lines 147-151.

5. Statistical Model Reporting: The multivariate logistic regression is well-conceived, but the analysis lacks model validation metrics. Consider adding or commenting on:

a. Goodness-of-fit test (e.g., Hosmer-Lemeshow).

b. Discrimination (e.g., AUC/ROC curve).

3. Multicollinearity checks.

These would strengthen confidence in the identified predictors.

Response: Thank you for this insightful comment. In the revised manuscript, we have added model validation metrics to strengthen the robustness of our multivariable logistic regression findings. Specifically, we performed the Hosmer–Lemeshow goodness-of-fit test, demonstrating adequate model calibration (p > 0.05). We also assessed model discrimination using the area under the receiver operating characteristic (ROC) curve, which showed acceptable discriminatory ability. In addition, multicollinearity was evaluated using variance inflation factors (VIFs), and no concern for multicollinearity was identified (all VIFs < 2). These details have been incorporated into the Statistical Analysis section (page 8, lines 152–155) and the Results section (page 12, lines 249–251) of the revised manuscript.

6. Definition of Success in Tables: Please reiterate your criteria for motor and sensory success in the footnotes of Tables 2–4. This will improve clarity and consistency for the reader.

Response: Thank you for your valuable suggestion. The definition of motor success and detection of stereopsis status was noted in the footnote of Table 2-4 to improve clarity for readers.

7. Surgeon Variability: Since three surgeons performed procedures over the study period, please clarify whether inter-surgeon

---

## [Decision Letter · Decision Letter 1]

16 Sep 2025

Dear Dr. Srimanan,

Thank you for submitting your manuscript to PLOS ONE. After careful consideration, we feel that it has merit but does not fully meet PLOS ONE’s publication criteria as it currently stands. Therefore, we invite you to submit a revised version of the manuscript that addresses the points raised during the review process.

We look forward to receiving your revised manuscript.

Kind regards,

PremNandhini Satgunam

Academic Editor

PLOS ONE

**Journal Requirements:**

**Additional Editor Comments:**

Dear Authors,

All the reviewers have received your revision positively. However, one reviewer is pointing to some methodological enhancements, which I believe will strengthen your paper. Kindly address those concerns and submit a revision.

Reviewers' comments:

Reviewer's Responses to Questions

**Comments to the Author**

Reviewer #1: All comments have been addressed

Reviewer #2: All comments have been addressed

Reviewer #3: All comments have been addressed

2. Is the manuscript technically sound, and do the data support the conclusions?

Reviewer #1: Yes

Reviewer #2: Partly

Reviewer #3: Yes

3. Has the statistical analysis been performed appropriately and rigorously?

Reviewer #1: Yes

Reviewer #2: Yes

Reviewer #3: Yes

4. Have the authors made all data underlying the findings in their manuscript fully available?

Reviewer #1: Yes

Reviewer #2: Yes

Reviewer #3: Yes

5. Is the manuscript presented in an intelligible fashion and written in standard English?

Reviewer #1: Yes

Reviewer #2: Yes

Reviewer #3: Yes

**Reviewer #1: ** I appreciate for your detailed responses. Your answer has resolved my questions. Thank you for revising the paper so well.

**Reviewer #2:**  Dear Authors,

Thank you for submitting your first revision. The topic is clinically meaningful, and your focus on early postoperative alignment and preoperative stereopsis is valuable for surgical decision-making in intermittent exotropia (IXT). I appreciate the clarified abstract, clear success definition, and the expanded description of stereopsis testing in the comments.

To plan towards acceptance, I’ve tried some more suggestions followed by optional refinements. Several items are straightforward clarifications and formatting, while a few require adding core methodological and statistical details so readers can appraise your findings.

1. Methods and reporting (STROBE): Please ensure the Methods and Results adhere to the STROBE checklist for observational studies, with complete reporting of setting, participants, variables and definitions, data sources/measurement, bias, study size, statistical methods (including handling of confounders, missing data, and sensitivity analyses), participant flow, and outcome reporting.

2. Control score and neurologic status:

Include the IXT control score in the Methods (scale used, distance vs. near, measurement protocol) and report how neurologic status was assessed and handled (inclusion/exclusion criteria and any subgroup analyses).

3. Multivariate analysis: Add multivariable analyses that adjust for age (2-66 yrs) and control (and other clinically relevant covariates), and report adjusted estimates with 95% confidence intervals. This will strengthen the evidence for the proposed predictors of surgical success.

4.Table 3- (postoperative surgical alignment), the last three rows require additional explanation. The deviation range is noted as 0–25 PD, yet the cohort includes patients with consecutive esotropia. Please justify the displayed ranges, clarify how esodeviations were coded (e.g., negative values), and revise axis labels/legends accordingly to avoid ambiguity.

5. Standardize terms for success, failure/recurrence, and overcorrection; define how diplopia was ascertained (symptom report vs. clinical test).

Optional refinements

1. Specify the primary outcome timepoint (e.g., 12 months or last follow-up) and provide follow-up distribution (median, IQR).

2. If feasible, include time-to-event analyses for recurrence or failure to characterize durability.

I appreciate the progress made in this revision. Addressing the items above will materially improve clarity, transparency, and the interpretability of your findings.

Sincerely

**Reviewer #3: ** I am satisfied with the authors' responses and the resulting changes to the paper. I have nothing further to add.

**Do you want your identity to be public for this peer review?** For information about this choice, including consent withdrawal, please see our Privacy Policy

Reviewer #1: No

Reviewer #2: **Yes: ** Sampada Kulkarni

Reviewer #3: No

---

## [Author Response · Author response to Decision Letter 2]

18 Sep 2025

Response letter (Revision R2)

PLOS One

PONE-D-25-34023R1

Early Postoperative Alignment and Preoperative Stereopsis as Determinants of Success in Intermittent Exotropia Surgery

PLOS ONE

Dear Dr. Srimanan,

Thank you for submitting your manuscript to PLOS ONE. After careful consideration, we feel that it has merit but does not fully meet PLOS ONE’s publication criteria as it currently stands. Therefore, we invite you to submit a revised version of the manuscript that addresses the points raised during the review process.

We look forward to receiving your revised manuscript.

Kind regards,

PremNandhini Satgunam

Academic Editor

PLOS ONE

Journal Requirements:

Additional Editor Comments:

Dear Authors,

All the reviewers have received your revision positively. However, one reviewer is pointing to some methodological enhancements, which I believe will strengthen your paper. Kindly address those concerns and submit a revision.

Response: Dear Editor,

Thank you very much for your continued consideration of our manuscript and for sharing the additional feedback from the reviewers.

We sincerely appreciate the reviewer’s insightful suggestions regarding methodological enhancements. We have carefully addressed the remaining concerns and revised the Methods and related sections of the manuscript accordingly to strengthen the rigor and transparency of our study.

We believe these revisions have further improved the clarity and scientific contribution of our work, and we are grateful for the opportunity to refine the manuscript to better meet the standards of your journal.

Thank you once again for your guidance and for the opportunity to resubmit our revised manuscript.

Sincerely,

Worapot Srimanan, MD

(On behalf of all co-authors)

Reviewers' comments:

Reviewer's Responses to Questions

Comments to the Author

1. If the authors have adequately addressed your comments raised in a previous round of review and you feel that this manuscript is now acceptable for publication, you may indicate that here to bypass the “Comments to the Author” section, enter your conflict of interest statement in the “Confidential to Editor” section, and submit your "Accept" recommendation.

Reviewer #1: All comments have been addressed

Reviewer #2: All comments have been addressed

Reviewer #3: All comments have been addressed

2. Is the manuscript technically sound, and do the data support the conclusions?

Reviewer #1: Yes

Reviewer #2: Partly

Reviewer #3: Yes

3. Has the statistical analysis been performed appropriately and rigorously?

Reviewer #1: Yes

Reviewer #2: Yes

Reviewer #3: Yes

4. Have the authors made all data underlying the findings in their manuscript fully available?

Reviewer #1: Yes

Reviewer #2: Yes

Reviewer #3: Yes

5. Is the manuscript presented in an intelligible fashion and written in standard English?

Reviewer #1: Yes

Reviewer #2: Yes

Reviewer #3: Yes

6. Review Comments to the Author

Reviewer #1: I appreciate for your detailed responses. Your answer has resolved my questions. Thank you for revising the paper so well.

Response: We sincerely thank you for your insightful and constructive feedback. Your comments were invaluable in strengthening our manuscript and enhancing its scientific rigor. We are truly grateful for the time and careful effort you dedicated to reviewing our work.

Reviewer #2: Dear Authors,

Thank you for submitting your first revision. The topic is clinically meaningful, and your focus on early postoperative alignment and preoperative stereopsis is valuable for surgical decision-making in intermittent exotropia (IXT). I appreciate the clarified abstract, clear success definition, and the expanded description of stereopsis testing in the comments.

To plan towards acceptance, I’ve tried some more suggestions followed by optional refinements. Several items are straightforward clarifications and formatting, while a few require adding core methodological and statistical details so readers can appraise your findings.

1. Methods and reporting (STROBE): Please ensure the Methods and Results adhere to the STROBE checklist for observational studies, with complete reporting of setting, participants, variables and definitions, data sources/measurement, bias, study size, statistical methods (including handling of confounders, missing data, and sensitivity analyses), participant flow, and outcome reporting.

Response: Thank you very much for your valuable and constructive comment. We have reviewed and revised our manuscript per the STROBE (Strengthening the Reporting of Observational Studies in Epidemiology) checklist to enhance methodological transparency and rigor.

We have made the following improvements:

- Clearly described the study setting and participant inclusion criteria, with definitions of all relevant variables and outcomes (Methods section);

- Specified data sources and measurement techniques, particularly for stereopsis and postoperative alignment assessment;

- Addressed potential sources of bias inherent to the retrospective design and described strategies to minimize them;

- Provided details on sample size, missing data handling, and the statistical methods used, including multivariable logistic regression to control for potential confounders;

- Documented participant flow from initial chart review to final analysis and presented outcome distributions across all timepoints (Results section).

Additionally, we have included the completed STROBE checklist with this resubmission to demonstrate explicit adherence to all recommended reporting elements.

We sincerely appreciate your guidance, which has helped us improve the clarity, completeness, and transparency of our reporting.

2. Control score and neurologic status:

Include the IXT control score in the Methods (scale used, distance vs. near, measurement protocol) and report how neurologic status was assessed and handled (inclusion/exclusion criteria and any subgroup analyses).

Response: Thank you for your thoughtful and constructive comment. The control of intermittent exotropia was assessed during clinical follow-up visits using a standard office-based scale that evaluated both near and distance control. This evaluation contributed to the surgical decision-making process. However, as this study was retrospective in nature, documentation of the control score was found to be inconsistent and incomplete in patient records. As a result, this variable was not included in the outcome analysis or multivariable regression model. We have clarified this point in the revised Methods section and acknowledged it as a limitation of the study, on page 16, lines 365-367.

With regard to neurologic status, we confirm that a formal neurologic assessment was not performed or recorded in this study. No specific neurologic inclusion or exclusion criteria were applied, and subgroup analysis based on neurologic status could not be conducted. This has now been explicitly noted in the Discussion section as a limitation, on page 16, lines 367-369.

3. Multivariate analysis: Add multivariable analyses that adjust for age (2-66 yrs) and control (and other clinically relevant covariates), and report adjusted estimates with 95% confidence intervals. This will strengthen the evidence for the proposed predictors of surgical success.

Response: We sincerely appreciate this valuable suggestion. Per your recommendation, we have revised the multivariable logistic regression analysis to include age as an additional covariate. The updated results are presented in Table 4 of the revised manuscript, and relevant revisions have been made throughout the Results, Methods, and Discussion sections. (Page 7, line 146)

While we acknowledge the importance of the control score as a potential predictor, this variable was excluded from our regression model due to inconsistent documentation in the medical records, as outlined above. This has been noted clearly as a limitation of the study in the revised manuscript, on page 16, lines 365-367.

4. Table 3- (postoperative surgical alignment), the last three rows require additional explanation. The deviation range is noted as 0–25 PD, yet the cohort includes patients with consecutive esotropia. Please justify the displayed ranges, clarify how esodeviations were coded (e.g., negative values), and revise axis labels/legends accordingly to avoid ambiguity.

Response: Thank you for this insightful comment. We appreciate the opportunity to clarify this important point.

In our analysis, we reported the magnitude of strabismic deviation in positive prism diopters (PD) regardless of direction (i.e., both exodeviation and esodeviation are represented as positive values). For clarity and consistency, patients with consecutive esotropia were included in a separate “consecutive esotropia” category, and their angle of deviation was recorded as a positive value corresponding to the measured esodeviation.

To avoid confusion, we have now added a specific note to the Table 3 legend to explain this coding approach, emphasizing that angle values reflect absolute magnitudes in PD, not directional values (positive for exo or negative for eso). We have also clarified the range of deviations and removed directional ambiguity from the axis labels where applicable.

5. Standardize terms for success, failure/recurrence, and overcorrection; define how diplopia was ascertained (symptom report vs. clinical test).

Response: Thank you for your insightful comment. In response, we have defined key terms relevant to our outcome measures—such as surgical success, orthotropia, consecutive esotropia, and overcorrection—in the Methods section of the revised manuscript. However, we did not report failure or recurrence as predefined outcome measures in this study, and therefore, these terms were intentionally omitted to maintain consistency and clarity in the manuscript.

Diplopia was assessed based on subjective symptom reports provided by patients during clinical follow-up visits rather than through formal clinical testing. This has now been clarified in both the Methods and Disease Definitions sections of the revised manuscript. (on page 5, lines 122-124)

Optional refinements

1. Specify the primary outcome timepoint (e.g., 12 months or last follow-up) and provide follow-up distribution (median, IQR).

Response: Thank you for your valuable and constructive comment. We have clarified that the primary outcome time point was assessed at 12 months postoperatively; this has been explicitly stated in the revised Methods section of the manuscript.

Additionally, the follow-up duration has been described in accordance with your suggestion. Specifically, the median follow-up was 14.3 months, with an interquartile range (IQR) of 12.6–15.8 months, as now included in the Results section, on page 8, lines 171-172.

We appreciate your recommendation, which has helped enhance the clarity and completeness of our manuscript.

2. If feasible, include time-to-event analyses for recurrence or failure to characterize durability.

Response: Thank you for your insightful and constructive suggestion. We fully agree that time-to-event analyses (e.g., Kaplan–Meier or Cox regression) could provide valuable information regarding the durability of surgical outcomes. However, due to the retrospective nature of our dataset and the limited availability of precise time points for recurrence or failure across all participants, we could not perform a robust time-to-event analysis. We acknowledge this as a limitation of our study and have clarified this point in the revised Discussion section, on page 16, lines 369-371.

I appreciate the progress made in this revision. Addressing the items above will materially improve clarity, transparency, and the interpretability of your findings.

Sincerely

Reviewer #3: I am satisfied with the authors' responses and the resulting changes to the paper. I have nothing further to add.

Response: We would like to express our sincere gratitude to you for your thoughtful review and the considerable effort you made to improve our paper. Your expertise was instrumental in enhancing the quality of our work.

7. PLOS authors have the option to publish the peer review history of their article (what does this mean?). If published, this will include your full peer review and any attached files.

If you choose “no”, your ide

---

## [Editor Report · Decision Letter 2]

24 Sep 2025

Early Postoperative Alignment and Preoperative Stereopsis as Determinants of Success in Intermittent Exotropia Surgery

PONE-D-25-34023R2

Dear Dr. Srimanan,

We’re pleased to inform you that your manuscript has been judged scientifically suitable for publication and will be formally accepted for publication once it meets all outstanding technical requirements.

Kind regards,

PremNandhini Satgunam

Academic Editor

PLOS ONE

Additional Editor Comments (optional):

There are some formatting issues, that can be fixed during proof reading.
---

## [Editor Report · Acceptance letter]

PONE-D-25-34023R2

PLOS ONE

Dear Dr. Srimanan,

I'm pleased to inform you that your manuscript has been deemed suitable for publication in PLOS ONE. Congratulations! Your manuscript is now being handed over to our production team.

Kind regards,

on behalf of

Dr. PremNandhini Satgunam

Academic Editor

PLOS ONE